# Impact of Decorin on the Physical Function and Prognosis of Patients with Hepatocellular Carcinoma

**DOI:** 10.3390/jcm9040936

**Published:** 2020-03-28

**Authors:** Takumi Kawaguchi, Sachiyo Yoshio, Yuzuru Sakamoto, Ryuki Hashida, Shunji Koya, Keisuke Hirota, Dan Nakano, Sakura Yamamura, Takashi Niizeki, Hiroo Matsuse, Takuji Torimura

**Affiliations:** 1Division of Gastroenterology, Department of Medicine, Kurume University School of Medicine, Kurume 830-0011, Japan; nakano_dan@med.kurume-u.ac.jp (D.N.); yamamura_sakura@med.kurume-u.ac.jp (S.Y.); niizeki_takashi@kurume-u.ac.jp (T.N.); tori@med.kurume-u.ac.jp (T.T.); 2Department of Liver Disease, Research Center for Hepatitis and Immunology, National Center for Global Health and Medicine, Kohnodai, Ichikawa 272-8516, Japan; sachiyo@hospk.ncgm.go.jp (S.Y.); yuzurusakamoto18@gmail.com (Y.S.); 3Department of Gastoenterological Surgery I, Hokkaido University Graduate School of Medicine, Sapporo 060-8638, Japan; 4Department of Orthopedics, School of Medicine, Kurume University, Kurume 830-0011, Japan; hashida_ryuuki@med.kurume-u.ac.jp (R.H.); matsuse_hiroh@kurume-u.ac.jp (H.M.); 5Division of Rehabilitation, Kurume University Hospital, Kurume 830-0011, Japan; kouya_shunji@kurume-u.ac.jp (S.K.); hirota_keisuke@kurume-u.ac.jp (K.H.)

**Keywords:** hepatoma, myokine, decorin, walking distance, survival

## Abstract

The outcome of patients with hepatocellular carcinoma (HCC) is still poor. Decorin is a small leucine-rich proteoglycan, which exerts antiproliferative and antiangiogenic properties in vitro. We aimed to investigate the associations of decorin with physical function and prognosis in patients with HCC. We enrolled 65 patients with HCC treated with transcatheter arterial chemoembolization (median age, 75 years; female/male, 25/40). Serum decorin levels were measured using enzyme-linked immunosorbent assays; patients were classified into the High or Low decorin groups by median levels. Associations of decorin with physical function and prognosis were evaluated by multivariate correlation and Cox regression analyses, respectively. Age and skeletal muscle indices were not significantly different between the High and Low decorin groups. In the High decorin group, the 6-min walking distance was significantly longer than the Low decorin group and was significantly correlated with serum decorin levels (*r* = 0.2927, *p* = 0.0353). In multivariate analysis, the High decorin group was independently associated with overall survival (hazard ratio 2.808, 95% confidence interval 1.016–8.018, *p* = 0.0498). In the High decorin group, overall survival rate was significantly higher than in the Low decorin group (median 732 days vs. 463 days, *p* = 0.010). In conclusion, decorin may be associated with physical function and prognosis in patients with HCC.

## 1. Introduction

Hepatocellular carcinoma (HCC) is a common cancer and the fourth leading cause of death due to cancer worldwide [1]. The incidence of HCC is predicted to continuously increase in both sexes and all age groups, since risk factors for HCC such as obesity, non-alcoholic steatohepatitis, and type 2 diabetes mellitus have become more prevalent worldwide [2]. In addition, the mortality rate of HCC has increased since 2000 [3], although there has been remarkable progresses in treatment for HCC, including the use of tyrosine kinase inhibitors [4]. The age-adjusted incidence and mortality rates of HCC are reported to be the highest in Eastern Asia [2]. The average 5-year survival rate is less than 15% in patients with HCC [5]. Thus, the prognosis of patients with HCC remains poor.

Skeletal muscle mass is known to be associated with the prognosis of patients with HCC [6]. Muscle atrophy is an independent factor associated with poor prognosis in patients with HCC treated with surgical resection and radiofrequency ablation [7]. Muscle atrophy is also a prognostic factor in patients with HCC treated with transarterial chemoembolization (TACE) and sorafenib [8,9]. In addition, muscle atrophy is associated with treatment tolerability and additional or subsequent therapies in patients with HCC treated with sorafenib [10]. In contrast, physical activity is associated with a reduced risk of HCC [11]. Moreover, exercise is reported to improve the prognosis of patients with HCC, regardless of changes in skeletal muscle mass [12].

Skeletal muscle is known as an endocrine organ [13]. By muscle contraction, myocytes release small peptides and cytokines, called myokines, and regulate muscle mass [13]. Myostatin is a myokine, which suppresses skeletal muscle growth and causes muscle atrophy [14]. Meanwhile, decorin is an exercise-induced myokine that suppresses muscle atrophy via inhibition of myostatin [15]. We previously reported that serum decorin levels are positively correlated with skeletal muscle mass in patients with HCC [16]. Decorin is also reported to interact with transforming growth factor-β and receptors of tyrosine kinase such as epidermal and insulin-like growth factors [17], leading to suppression of proliferation of various tumor cell lines, including HCC cell lines [18,19,20]. In addition, decorin is known to be expressed in various tissues including intestinal tissue, cardiac tissue, and adipose tissue and is known to regulate autophagy, inflammation, and glucose homeostasis [21,22,23,24]. Thus, accumulated evidence from basic studies suggests that decorin has an impact on the prognosis of patients with HCC. However, there has been no clinical study investigating the prognostic impact of decorin in patients with HCC.

The aim of this study was to investigate the association of serum decorin levels with physical function and prognosis in patients with HCC.

## 2. Materials and Methods

### 2.1. Study Design

This was a retrospective study to investigate the impact of serum decorin levels on the physical function and prognosis of patients with HCC.

### 2.2. Ethics

The study protocol conformed to the ethical guidelines of the Declaration of Helsinki and was approved by the institutional review board of Kurume University. We employed an opt-out approach to obtain informed consent from patients.

### 2.3. Subjects

We registered 339 consecutive patients with HCC between November 2014 and March 2018. Of these patients, 165 patients were excluded because of radiofrequency ablation (*n* = 43), hepatic arterial infusion chemotherapy (*n* = 91), tyrosine-kinase inhibitor (*n* = 23), or radiation (*n* = 8), and the remaining 174 patients with HCC who underwent TACE were selected. Of the 174 patients with HCC who underwent TACE, 105 patients were excluded because of hepatic encephalopathy (*n* = 27), HCC rupture (*n* = 17), renal failure (*n* = 7), or lack of data for physical function tests (*n* = 54). Finally, a total of 69 patients with HCC were analyzed in this study (Figure 1). We classified all patients into the High or Low decorin group per the median decorin level.

### 2.4. Diagnosis, Barcelona Clinic Liver Cancer (BCLC) Staging, and Treatment of HCC

HCC was diagnosed and treated according to the guidelines for HCC of the Japan Society of Hepatology [25]. The clinical stage of HCC was evaluated using the BCLC staging system [26].

### 2.5. Measurement of Skeletal Muscle Index (SMI) and Visceral Fat Area

The SMI was evaluated using computed tomography (CT) images obtained at the diagnosis of HCC as previously described [27,28]. The skeletal muscle mass was measured by manual tracings on CT images, and their sum was calculated using ImageJ Version 1.50 software (National Institutes of Health, Bethesda, MD, USA) [29]. The skeletal muscle mass was evaluated by the SMI.

### 2.6. Measurement of Physical Function 

Grip strength and the 6-min walking distance were evaluated by qualified physical therapists. Handgrip was measured on the non-dominant hand using a dynamometer (TKK5401; Takei Scientific Instruments Co., Ltd., Niigata, Japan) [6]. The 6-minute walking distance was measured by evaluating the total ambulated distance [30].

### 2.7. Diagnosis of Sarcopenia

The diagnosis of sarcopenia was based on the Japan Society of Hepatology diagnostic criteria for sarcopenia in patients with liver disease [6]. Patients who showed both a decrease in grip strength (the cut-off value is 26 kg for men and 18 kg for women) and a decrease in skeletal muscle mass (the cut-off value of SMI is 42 cm^2^/m^2^ for men and 38 cm^2^/m^2^ for women) were diagnosed with sarcopenia. The other patients were classified as non-sarcopenia [6].

### 2.8. Biochemical Tests

Blood samples were obtained at the baseline in the early morning after an overnight fast. The blood biochemical tests performed were for serum levels of alpha-fetoprotein, des-γ-carboxy prothrombin, liver function tests, renal function tests, total cholesterol, creatine kinase, and hemoglobin A1c. We also measured the complete blood cell count.

### 2.9. Measurement of Serum Levels of Myostatin, FGF-21 and Decorin

Serum levels of myostatin, FGF-21, and decorin were measured using a Myostatin Quantikine enzyme-linked immunosorbent assay (ELISA) Kit (R&D Systems, Inc., Minneapolis, MN, USA), Human FGF-21 ELISA Kit (BioVendor—Laboratorni medicina a.s., Brno, Czech Republic), and Human Decorin ELISA Kit (Abcam plc., Cambridge, UK) according to the manufacturers’ instructions, respectively.

### 2.10. Follow-Up and Definition of Survival Term

After treatment with TACE, patients were followed up until death or the study censor date through routine physical examinations, biochemical tests, and abdominal imaging including ultrasonography, CT, or magnetic resonance imaging according to the HCC guidelines of the Japan Society of Hepatology [25]. The median observational period was 617 days (range, 52–2068 days). The survival term was defined as the period from the diagnosis of HCC to death or the study censor date.

### 2.11. Statistical Analysis

Data are expressed as the median (interquartile range), range, or number. The differences between the High and Low decorin groups were analyzed using Wilcoxon rank sum tests. Factors correlated with serum decorin levels were evaluated by pairwise correlations [31]. In addition, independent factors associated with survival were analyzed using Cox regression analysis, as previously described [27]. The overall survival in the High and Low decorin groups was estimated using the Kaplan–Meier method, and differences in survival between the groups were analyzed using the log-rank test. All the statistical analyses were performed using JMP Pro^®^ 14 (SAS Institute Inc., Cary, NC, USA). Values of *p* < 0.05 were considered to indicate statistically significant differences.

## 3. Results

### 3.1. Patient Characteristics

The patient characteristics are summarized in Table 1. There was no significant difference in age and body mass index. The prevalence of men was significantly higher in the High decorin group than that in the Low decorin group. There was no significant difference in the hospitalization period between the two groups.

In the High decorin group, the prevalence of sarcopenia was significantly lower than that in the Low decorin group. Although no significant differences were noted in the SMI and serum creatine kinase level between the two groups, the 6-min walking distance in the High decorin group was significantly longer than that in the Low decorin group (Table 1).

Although there was no significant difference in the serum fibroblast growth factor (FGF)-21 level between the two groups, the serum level of myostatin was significantly higher in the High decorin group than that in the Low decorin group (Table 1).

There was no significant difference in the BCLC classification between the High and Low decorin groups. No significant difference in the serum alpha-fetoprotein level was also observed between the two groups; however, in the High decorin group, the serum des-γ-carboxy prothrombin level was significantly lower than that in the Low decorin group (Table 1).

There was no significant difference in the prevalence of Child–Pugh class B and use of branched-chain amino acid supplementation between the two groups. Serum levels of aspartate aminotransferase, alanine aminotransferase, and estimated glomerular filtration rate were significantly higher in the High decorin group than those in the Low decorin group. In the High decorin group, serum levels of blood urea nitrogen, creatinine, and triglycerides and the hemoglobin A1c value were significantly lower than those in the Low decorin group (Table 1).

### 3.2. Multivariate Correlation Analysis Between Serum Decorin Levels and Each Variable

No significant correlation was seen between serum decorin levels and age, body mass index, grip strength, SMI, levels of alpha-fetoprotein, albumin, total bilirubin, creatine kinase, hemoglobin A1c, and estimated glomerular filtration rate. Serum decorin levels showed a significant negative correlation with serum des-γ-carboxy prothrombin levels. Serum decorin levels demonstrated a significant positive correlation between the 6-min walking distance and serum myostatin levels (Table 2).

### 3.3. Independent Factors Associated with Survival

We examined independent factors associated with survival and found that high decorin levels were identified as an independent factor of better overall survival. Meanwhile, the BCLC stage and Child–Pugh class were not identified as independent factors associated with overall survival (Table 3).

### 3.4. Kaplan–Meier Analysis for Survival

In the High decorin group, the overall survival rate was significantly higher compared to that in the Low decorin group (median 732 days vs. 463 days; log-rank test *p* = 0.0498) (Figure 2A). In the subgroup analysis of BCLC stage B, the difference in overall survival rates between the High and Low decorin groups became more significant than that in the analysis in all subjects (BCLC stages A and B) (Figure 2B).

## 4. Discussion

In this study, we demonstrated that serum decorin levels were positively correlated with the 6-min walking distance, an index of cardiopulmonary function in patients with HCC. In addition, we found that serum decorin levels were an independent prognostic factor in patients with HCC. Although more research is needed and our data are preliminary in essence, these data suggest that decorin may be associated with physical function and prognosis in patients with HCC.

TACE is a standard treatment for intermediate-stage HCC [26,32]. In this study, we enrolled patients with HCC treated with TACE, and the median survival period was 617 days, which is comparable to that reported previously [26,33]. The prognosis of patients with HCC is dependent on the BCLC stage [26]. However, the BCLC stage was not identified as an independent prognostic factor in this study, and the reason for this remains unclear. However, all enrolled patients with HCC were treated with TACE, and patients with the BCLC stage B accounted for about 90% of the enrolled patients. Therefore, the narrow distribution of the BCLC stage may be a possible explanation.

Although myostatin and FGF-21 are myokines, the levels of these myokines were not identified as independent prognostic factors in patients with HCC. Nishikawa et al. reported that elevated serum myostatin levels are associated with worse survival in patients with liver cirrhosis [34]. Hyperammonemia has been reported to transcriptionally upregulate myostatin through nuclear transport of p65 nuclear factor-ƙB, resulting in sarcopenia and poor prognosis [35]. Meanwhile, patients with hepatic encephalopathy (West Haven criteria grade II–IV) were excluded, and the prevalence of hyperammonemia was thought to be low in this study. Therefore, myostatin may not be identified as a prognostic factor. Deficiency of FGF-21 is reported to promote HCC in mice receiving a long-term obesogenic diet [36]. Long-term administration of FGF-21 prevents chemically induced hepatocarcinogenesis in mice [37]. However, FGF-21 is known to be expressed in several tissues, including those of the liver, fat, and pancreas [38]. Serum FGF-21 levels are affected by various tissues expressing FGF-21, and, therefore, FGF-21 was not identified as an independent prognostic factor in patients with HCC.

Serum decorin levels were positively correlated with the 6-min walking distance, an index of cardiopulmonary function in patients with HCC. Overexpression of decorin is reported to ameliorate diabetic cardiomyopathy and cardiac function in rats [39]. N-terminal cleavage of decorin confers an inhibitory effect against myostatin, suppressing the atrophy of cardiomyocytes [40]. In fact, serum decorin level was positively correlated with serum myostatin level in this study. One would think that decorin may be up regulated to suppress muscle atrophy in response to an increase in serum myostatin level. In addition, C-terminal truncation of decorin interacts with the connective tissue growth factor, leading to suppression of myocardial fibrosis through down-regulation of cardiac extracellular matrix production [40]. Furthermore, Kwon et al. reported that decorin causes macrophage polarization via cluster of differentiation-44, resulting in an amelioration of pulmonary function in a rat model of hypertoxic lung damage [41]. These previous basic studies, along with our results, may suggest that decorin may be associated with cardiopulmonary function in patients with HCC (Figure 3). However, the correlation between serum decorin level and the 6-min walking distance could not lead to the conclusion that the high decorin level is a cause of high cardiovascular fitness, in such a small number of subjects.

In this study, we first examined the impact of the serum decorin level in patients with HCC and found that serum decorin levels were identified as an independent prognostic factor in patients with HCC. Moreover, in the stratification analysis according to the BCLC stage, the prognostic impact of decorin was more evident in patients with HCC with the BCLC stage B. Horváth et al. reported that genetic ablation of decorin leads to enhanced hepatocarcinogenesis compared to that in wild-type animals [42]. Meanwhile, recombinant human decorin inhibits the proliferation of HepG2 cells [43,44]. Several mechanisms for decorin-induced inhibition of cell proliferation have been reported. Decorin is reported to reduce the secretion of transforming growth factor-β1 in HCC cell lines [20]. Decorin is also reported to downregulate the phosphorylation of epidermal growth factor receptor, glycogen synthase kinase 3β, and extracellular signal-regulated kinase 1/2 [20]. In addition, decorin suppresses the ATR/Chk1/Wee1 axis, leading to inhibition of the cell cycle in the G2/M phase via phosphorylation of cyclin-dependent kinase 1 [20]. Moreover, decorin is known to decrease the expression of pro-angiogenic factors, vascular endothelial growth factor A, and hypoxia-inducible factor 1-α, resulting in downregulation of the hepatocyte growth factor and epidermal growth factor receptor signaling axes [45]. In fact, the serum decorin level was negatively correlated with the serum des-γ-carboxy prothrombin level, which is a tumor maker for HCC in this study. Thus, decorin may suppress the proliferation of HCC through direct and indirect tumor inhibitory effects and may be associated with prognosis in patients with HCC (Figure 3). However, decorin is known to be expressed not only in skeletal muscle [15], but also in various tissues including intestinal tissue, cardiac tissue, and adipose tissue [21,22,23,24]. Accordingly, it remains unclear where decorin comes from in the present study (Figure 3). In addition, we have to be cautious of the interpretation of our data. Expression of decorin is recently reported to be seen in the tumor cell such as glioblastoma and is negatively associated with the overall survival rate of patients with glioblastoma multiforme [46]. Thus, further research is required to investigate the expression of decorin in HCC tissue and a causal relationship between decorin and prognosis of the patients with HCC.

Limitations of this study are the following: First, this was a retrospective study conducted in a single center. Second, the number of enrolled subjects is very limited to examine independent prognostic factors. Third, we enrolled patients with HCC treated with TACE. It remains unclear if serum decorin levels have a prognostic impact in patients with HCC treated with hepatic resection or tyrosine kinase inhibitors. Fourth, no patient underwent liver transplantation during the observation period, suggesting the selection bias. Thus, a multicenter prospective cohort study should be conducted with various HCC stages and treatments for HCC including liver transplantation.

## 5. Conclusions

In conclusion, we demonstrated that serum decorin levels were positively correlated with cardiopulmonary function in patients with HCC. In addition, serum decorin levels were an independent prognostic factor in patients with HCC. Although more research is needed and our data are preliminary in essence, the results of this study may suggest that decorin may be associated with physical function and prognosis in patients with HCC.

## Figures and Tables

**Figure 1 jcm-09-00936-f001:**
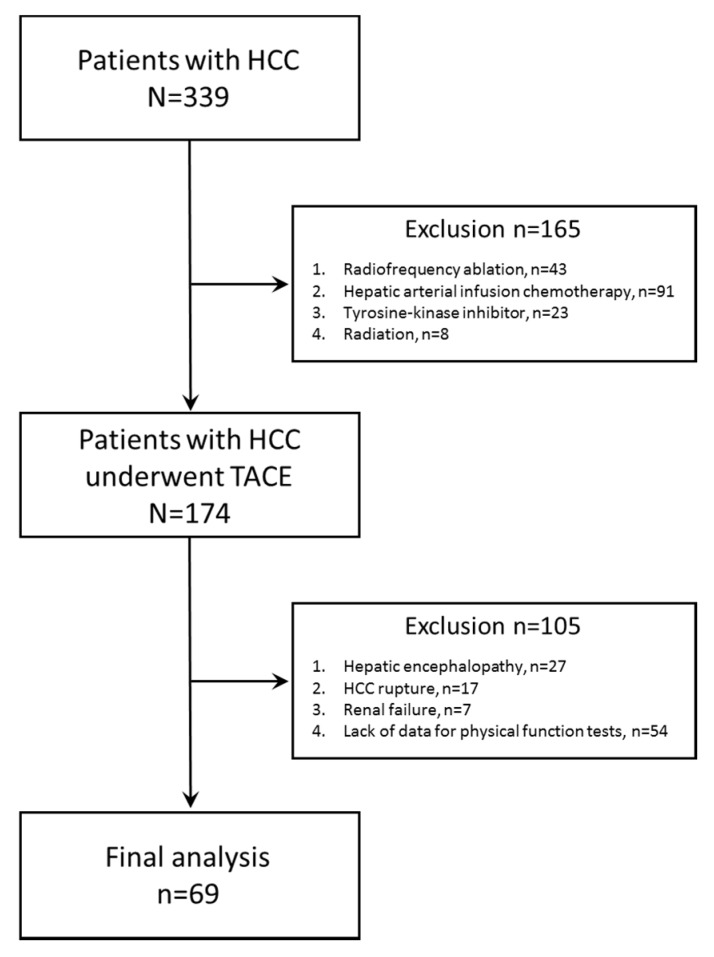
A flow diagram of analyzed subjects.

**Figure 2 jcm-09-00936-f002:**
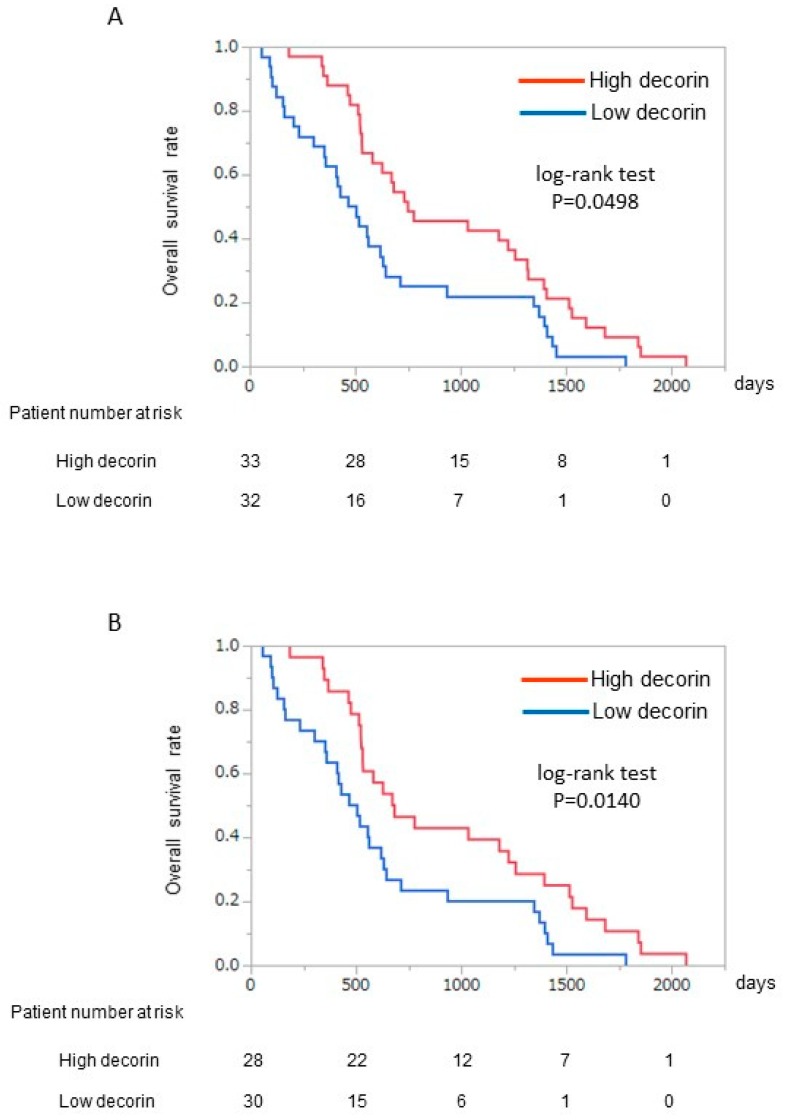
Kaplan–Meier analysis between the High decorin and Low decorin groups. (**A**) All patients; (**B**) Patients with BCLC stage B HCC. BCLC, Barcelona Clinic Liver Cancer; HCC, hepatocellular carcinoma.

**Figure 3 jcm-09-00936-f003:**
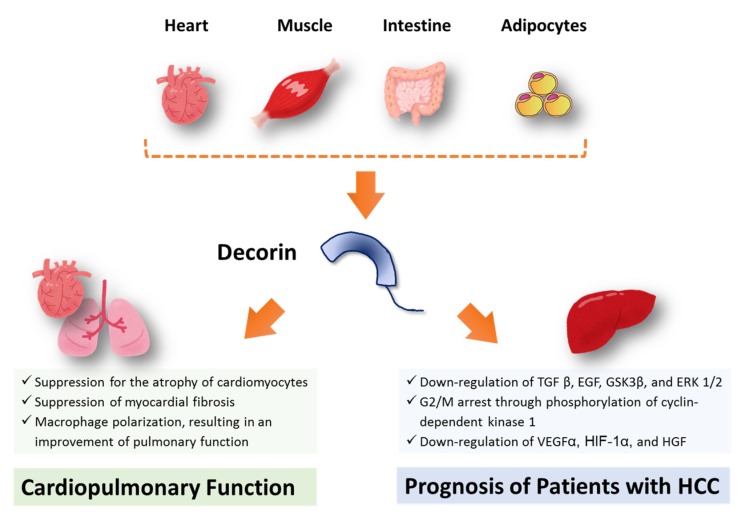
A scheme for the proposed hypothesis of this study. Decorin is expressed in various tissues including skeletal muscle, heart, intestine, and adipocytes. In this study, it remains unclear where decorin comes from. Decorin may be associated with cardiopulmonary function, because decorin suppresses the atrophy of cardiomyocytes, myocardial fibrosis, and causes macrophage polarization. In addition, decorin may be associated with prognosis of patients with HCC, because decorin downregulates transforming growth factor-β1, epidermal growth factor receptor, glycogen synthase kinase 3β, and extracellular signal-regulated kinase 1/2, G2/M arrest through phosphorylation of cyclin-dependent kinase 1, downregulation of vascular endothelial growth factor A, hypoxia-inducible factor 1-α, and hepatocyte growth factor. Abbreviations: TGF β, transforming growth factor-β1; EGF, epidermal growth factor receptor; GSK3β, glycogen synthase kinase 3β; and ERK, extracellular signal-regulated kinase; VEGF, vascular endothelial growth factor; HIF-1α, hypoxia-inducible factor 1-α; HGF, hepatocyte growth factor.

**Table 1 jcm-09-00936-t001:** Patients’ characteristics.

	All Subjects		High Decorin		Low Decorin		
	Median (IQR)	Range (min–max)	Median (IQR)	Range (min–max)	Median (IQR)	Range (min–max)	*p*
Number (*n*)	65	N/A	33	N/A	32	N/A	N/A
Age (years)	75 (71–80)	60–90	76 (72–80)	60–89	75 (71–80)	63–90	0.9528
Sex (women/men)	38.5%/61.5% (25/40)	N/A	54.5%/45.5% (18/15)	N/A	21.9%/78.1% (7/25)	N/A	0.0068
Body mass index (kg/m^2^)	23.9 (21.2–26.0)	16.7–37.8	23.0 (21.5–26.3)	19.6–30.3	24.1 (20.5–25.8)	16.7–37.8	0.6227
Hospitalization period (days)	14 (11–21)	7–55	13 (11.5–17.5)	7–55	17 (11–21)	7–34	0.3076
Grip strength (kg)	24.2 (20.3–31.3)	13.3–42.8	24.1 (19.25–30.63)	13.3–42.8	25.0 (21.5–32.1)	14.9–39	0.8045
Skeletal muscle index (cm^2^/m^2^)	29.69 (23.94–35.20)	11.85–51.18	28.50 (23.21–34.22)	11.85–41.79	31.53 (24.33–36.82)	12.45–51.18	0.3758
Sarcopenia (Presence/Absence)	18.5%/81.5% (12/53)	N/A	6.1%/93.9% (2/31)	N/A	31.3%/68.7% (10/22)	N/A	0.0089
Visceral fat area (cm^2^)	61.7 (39.8–84.6)	4.4–240.8	59.2 (39.8–78.1)	24.2–240.8	61.9 (36.5–95.9)	4.4–197.8	0.7578
Serum creatine kinase (U/L)	95 (70.5–132.5)	17–374	99 (77–133)	44–246	90.5 (63.75–129.5)	17–374	0.3558
6-minute walking distance (m)	379 (302–420)	26–621	391 (365–433)	228–621	334 (255–407)	26–501	0.0093
Decorin (pg/mL)	17,322 (13,499–21,866)	7400–32,102	21,799 (18,990–27,033)	17,322–32,102	13,499 (11,939–14,726)	7400–16,838	<0.0001
Myostatin (pg/mL)	1699 (1180–3658)	245–7788	3056 (1313–4610)	501–6350	1500 (1154–3214)	246–7788	0.0426
FGF-21 (pg/mL)	160 (116–344)	13–2150	174 (140–340)	39–2150	156 (96–357)	13–1328	0.4197
BCLC stage (A/B)	10.8%/89.2% (7/58)	N/A	15.2%/84.8%(5/28)	N/A	6.2%/93.8%(2/30)	N/A	0.2471
AFP (ng/mL)	32.75 (6.83–275.18)	1.4–67,036	20 (8.25–72.27)	3.9–1594	93 (5.5–1613)	1.4–67,036	0.2318
DCP (mAU/mL)	76 (28–888.5)	9–30,844	36 (24–211.5)	12–17,353	144 (43–6753)	9–30,844	0.0253
Child–Pugh class (A/B)	69.2%/30.8% (45/20)	N/A	75.8%/24.2% (25/8)	N/A	62.5%/37.5% (20/12)	N/A	0.2469
BCAA supplementation (With/Without)	52.3%/47.7% (34/31)	N/A	51.5%/48.5% (17/16)	N/A	53.1%/46.9% (17/15)	N/A	0.8966
AST (IU/L)	43 (32–55.5)	19–158	45 (40–65.5)	23–158	34 (26–48)	19–99	0.0009
ALT (IU/L)	28 (21–37.5)	7–186	32 (24.5–41)	20–186	23 (17.5–32.5)	7–87	0.0031
ALP (IU/L)	351 (291–479)	180–854	356 (309.5–541.5)	200–854	325 (275.75–455)	180–659	0.2299
GGT (IU/L)	44 (26–73.5)	9–551	45 (26.5–79.5)	15–551	42 (26–66.5)	9–252	0.6088
Total protein (g/dL)	7.28 (6.72–7.78)	5.94–8.89	7.36 (6.62–7.81)	5.94–8.15	7.28 (6.82–7.62)	6.04–8.89	0.9477
Albumin (g/dL)	3.5 (3.1–3.7)	2.5–4.3	3.4 (3.1–3.7)	2.5–4.3	3.5 (3.0–3.8)	2.8–4.2	1.0000
Total bilirubin (mg/dL)	0.9 (0.6–1.3)	0.3–2.8	0.9 (0.6–1.3)	0.4–2.8	0.9 (0.6–1.2)	0.3–1.6	0.5996
Prothrombin activity (%)	80 (68–88)	38–117	81 (65.5–90)	42–117	79 (69–85.5)	38–108	0.6365
Blood urea nitrogen (mg/dL)	17 (14–19.6)	5.9–47.6	14.9 (13–18.9)	5.9–28.1	17.45 (15–20.18)	11.5–47.6	0.0253
Creatinine (mg/dL)	0.74 (0.61–0.92)	0.43–1.91	0.66 (0.55–0.77)	0.43–1.52	0.81 (0.66–1.03)	0.56–1.91	0.0013
eGFR (mL/min/1.73 m^2^)	73.2 (54.3–84.85)	27.3–121.3	78.6 (62.15–89.95)	34.3–121.3	65.7 (52.63–75.8)	27.3–102.4	0.0107
Total cholesterol (mg/dL)	144 (126–162)	79–233	138 (121–156)	84–197	147 (128–163)	79–233	0.3154
Triglyceride (mg/dL)	82 (70–108)	28–249	75 (64–94)	28–249	91 (78–130)	54–179	0.0237
HbA1c (%)	5.8 (5.5–6.4)	4.3–13.4	5.7 (5.25–6.1)	4.3–8.3	6.1 (5.7–6.8)	4.7–13.4	0.0268
Red blood cell count (×10^4^/µL)	389 (355–420)	249–615	385 (356–416)	310–455	393 (347–442)	249–615	0.7528
Hemoglobin (g/dL)	11.9 (10.4–12.75)	7.3–15.4	11.9 (10.6–12.8)	7.3–15.4	11.9 (9.8–12.7)	7.3–14.9	0.7083
White blood cell count (/µL)	3800 (3100–5050)	1800–7900	3700 (3050–5560)	1900–6600	4150 (3125–5525)	1800–7900	0.2270
Platelet count (×10^3^/mm^3^)	10.9 (8.35–15.1)	3.2–31.8	9.4 (7.8–13.0)	3.2–22.6	11.8 (8.6–16.1)	4.0–31.8	0.0881

Note: Data are expressed as median (interquartile range (IQR)), range, or number. Abbreviations: AFP, alpha-fetoprotein; ALP, alkaline phosphatase; ALT, alanine aminotransferase; AST, aspartate aminotransferase; BCAA, branched-chain amino acid; BCLC, Barcelona Clinic Liver Cancer; DCP, des-γ-carboxy prothrombin; eGFR, estimated glomerular filtration rate; FGF-21, fibroblast growth factor-21; GGT, gamma-glutamyl transpeptidase; HbA1c, hemoglobin A1c; N/A, not applicable

**Table 2 jcm-09-00936-t002:** Multivariate correlation analysis between serum decorin levels and each variable.

Variable	Correlation Coefficient	*p*
Age	−0.0250	0.8750
Body mass index	0.0415	0.7942
Grip strength	−0.0532	0.7380
Skeletal muscle index	−0.1362	0.3898
Visceral fat area	0.0278	0.861
6-min walking distance	0.2927	0.0353
Creatine kinase	−0.0062	0.9690
Myostatin	0.3200	0.0389
FGF-21	−0.0352	0.8249
AFP	−0.2270	0.1482
DCP	−0.3476	0.0241
AST	0.2453	0.0992
ALT	0.2734	0.0798
ALP	0.1260	0.4266
GGT	0.0042	0.979
Total protein	−0.0197	0.9015
Albumin	−0.1754	0.2664
Total bilirubin	0.1054	0.5063
Prothrombin activity	0.1078	0.4968
Blood urea nitrogen	−0.1606	0.3095
Creatinine	−0.1650	0.2965
eGFR	0.0695	0.6617
Total cholesterol	−0.0914	0.5650
Triglyceride	−0.0594	0.7089
HbA1c	−0.2748	0.0782
Red blood cell count	−0.1337	0.3984
Hemoglobin	−0.0384	0.8091
White blood cell count	−0.233	0.1376
Platelet count	−0.2261	0.15

Abbreviations: FGF-21, fibroblast growth factor-21; AFP, alpha-fetoprotein; DCP, des-γ-carboxy prothrombin; AST, aspartate aminotransferase; ALT, alanine aminotransferase; ALP, alkaline phosphatase; GGT, gamma-glutamyl transpeptidase; eGFR, estimated glomerular filtration rate; HbA1c, hemoglobin A1c.

**Table 3 jcm-09-00936-t003:** Multivariate Cox regression analysis for overall survival.

Factors	Hazard Ratio	95% Confidence Interval	*p*-Value
Decorin (High/Low)	2.808	1.016–8.018	0.0498
BCLC stage (A/B–C)	6.720	0.707–73.877	0.0553
Child–Pugh class (A/B)	1.436	0.461–4.473	0.5308

Abbreviations: BCLC, Barcelona Clinic Liver Cancer.

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
