# Peer review of "Impact of Decorin on the Physical Function and Prognosis of Patients with Hepatocellular Carcinoma"

_jcm, 2020, doi:10.3390/jcm9040936_

Round 1
Reviewer 1 Report
Kawaguchi T et al conducted a retrospective clinical study to evaluate the impact of decorin (myokine and metrykine) on physical function and prognosis of HCC, and suggested decorin as an "independent prognostic factor" of HCC. The manuscript has been well-written and provided a clinical evidence supporting that there is "some" association between serum decorin level and clinical parameters of HCC. However, there are several points that should be considered.
First, the number of subjects is very limited to conclude that decorin is an independent prognostic factor. In addition, authors did not provide the scientific interpretation of their results. It has been reported that decorin presents various physiological and pathological function, and is expressed in various tissues including tumor. Therefore, authors' interpretation seems to be seriously biased.
Second, authors augured that high decorin level is a cause of high cardiovascular fitness, however there is no results supporting this conclusion. The correlation between decorin and 6-min walking distance could not lead to the conclusion, in such a small number of subjects.
Finally, authors concluded that decorin can be a therapeutic target to improve prognosis of HCC. However, the current results is too weak to prove cause-effect relationship for such a strong conclusion.
In addition, several minor points should be improved, including detailed description of statistical methods for correction of multiple comparison, why authors did not discuss other positive results, and so on.
Author Response
To REVIEWER 1,
Thank you very much for your letter regarding our manuscript (jcm-734136). We appreciate your comments, which have helped us to improve our manuscript. In line with your comments, please find below our point-by-point responses.
- First, the number of subjects is very limited to conclude that decorin is an independent prognostic factor. In addition, authors did not provide the scientific interpretation of their results. It has been reported that decorin presents various physiological and pathological function, and is expressed in various tissues including tumor. Therefore, authors' interpretation seems to be seriously biased.
Response: As you pointed out, the number of subjects is very limited to conclude that decorin is an independent prognostic factor in this study. Since it is difficult to enroll sufficient number of patients at present, we have described this issue as a limitation of this study (line 279-280).
We also agree that our interpretation is seriously biased and we have to be cautious of interpretation of our data. Decorin is reported to be expressed in various tissues and present various physiological and pathological function. In addition, expression of decorin is reported to be seen in glioblastoma [1]. To reduce the interpretation bias, we revised the discussion as followings: Decorin is known to be expressed not only in skeletal muscle [2], but also in various tissues including intestinal tissue, cardiac tissue, and adipose tissue [3-6]. Accordingly, it remains unclear where decorin comes from in the present study. In addition, we have to be cautious of interpretation of our data. expression of decorin is recently reported to be seen in tumor cell such as glioblastoma and is negatively associated with the overall survival rate of patients with glioblastoma multiforme [1]. Thus, further research is required to investigate the expression of decorin in HCC tissue and a causal relationship between decorin and prognosis of the patients with HCC (line 256-263). Again, we appreciate your comment, which have helped us to improve our manuscript.
- Second, authors augured that high decorin level is a cause of high cardiovascular fitness, however there is no results supporting this conclusion. The correlation between decorin and 6-min walking distance could not lead to the conclusion, in such a small number of subjects.
Response: As you suggested, we did not present the results which support the causal relationship between high decorin level and high cardiovascular fitness. In the revised manuscript, we revised the description as followings: These previous basic studies, along with our results, may suggest that decorin may be associated with cardiopulmonary function in patients with HCC. However, the correlation between serum decorin level and 6-min walking distance could not lead to the conclusion that high decorin level is a cause of high cardiovascular fitness, in such a small number of subjects (line 234-238).
- Finally, authors concluded that decorin can be a therapeutic target to improve prognosis of HCC. However, the current results is too weak to prove cause-effect relationship for such a strong conclusion.
Response: As you pointed out, we did not present any result for causal relationship between an increase in serum decorin level and an improvement of prognosis of patients with HCC. Thus, it is difficult to conclude that decorin can be a therapeutic target to improve prognosis of HCC. In the revised manuscript, we have removed the description that decorin is a possible the therapeutic target from the manuscript and revised the conclusion as followings: Although more research is needed and our data are preliminary in essence, the results of this study may suggest that decorin may be associated with physical function and prognosis in patients with HCC (line 202-203, line 289-291).
- In addition, several minor points should be improved, including detailed description of statistical methods for correction of multiple comparison, why authors did not discuss other positive results, and so on.
Response: As you suggested, we did not discuss a significant correlation of serum decorin level with serum myostatin level. Since N-terminal cleavage of decorin confers an inhibitory effect against myostatin [7], one would think that decorin may be up-regulated to suppress muscle atrophy in response to an increase in serum myostatin level (line 227-230).
In addition, there was a significant negative correlation between serum decorin level and serum des-γ-carboxy prothrombin (DCP) level, which is a tumor marker for hepatocellular carcinoma. Decorin is reported to down-regulate cell proliferation through inhibition of transforming growth factor-β1, epidermal growth factor receptor, glycogen synthase kinase 3β, and extracellular signal-regulated kinase 1/2 [8]. In addition, decorin is known to suppress the cell cycle in the G2/M phase [8] and down-regulate the expression of pro-angiogenic factors [9]. Thus, decorin may exert anti-tumor effects though various mechanisms and, therefore, serum decorin level may be negatively correlated with serum DCP levels in this study (line 253-254). In the revised manuscript, we added these descriptions in the Discussion section.
References
1 Tsidulko, A.Y.; Kazanskaya, G.M.; Volkov, A.M.; Suhovskih, A.V.; Kiselev, R.S.; Kobozev, V.V.; Gaytan, A.S.; Krivoshapkin, A.L.; Aidagulova, S.V., Grigorieva, E.V. Chondroitin sulfate content and decorin expression in glioblastoma are associated with proliferative activity of glioma cells and disease prognosis. Cell Tissue Res. 2020; 379: 147-55.
2 Kanzleiter, T.; Rath, M.; Gorgens, S.W.; Jensen, J.; Tangen, D.S.; Kolnes, A.J.; Kolnes, K.J.; Lee, S.; Eckel, J.; Schurmann, A.; et al. The myokine decorin is regulated by contraction and involved in muscle hypertrophy. Biochem Biophys Res Commun. 2014; 450: 1089-94.
3 Svard, J.; Rost, T.H.; Sommervoll, C.E.N.; Haugen, C.; Gudbrandsen, O.A.; Mellgren, A.E.; Rodahl, E.; Ferno, J.; Dankel, S.N.; Sagen, J.V.; et al. Absence of the proteoglycan decorin reduces glucose tolerance in overfed male mice. Sci Rep. 2019; 9: 4614.
4 Zhao, H.; Xi, H.; Wei, B.; Cai, A.; Wang, T.; Wang, Y.; Zhao, X.; Song, Y., Chen, L. Expression of decorin in intestinal tissues of mice with inflammatory bowel disease and its correlation with autophagy. Exp Ther Med. 2016; 12: 3885-92.
5 Gubbiotti, M.A.; Neill, T.; Frey, H.; Schaefer, L., Iozzo, R.V. Decorin is an autophagy-inducible proteoglycan and is required for proper in vivo autophagy. Matrix Biol. 2015; 48: 14-25.
6 Pohle, T.; Altenburger, M.; Shahin, M.; Konturek, J.W.; Kresse, H., Domschke, W. Expression of decorin and biglycan in rat gastric tissue: effects of ulceration and basic fibroblast growth factor. Scand J Gastroenterol. 2001; 36: 683-9.
7 Barallobre-Barreiro, J.; Gupta, S.K.; Zoccarato, A.; Kitazume-Taneike, R.; Fava, M.; Yin, X.; Werner, T.; Hirt, M.N.; Zampetaki, A.; Viviano, A.; et al. Glycoproteomics Reveals Decorin Peptides With Anti-Myostatin Activity in Human Atrial Fibrillation. Circulation. 2016; 134: 817-32.
8 Horvath, Z.; Reszegi, A.; Szilak, L.; Danko, T.; Kovalszky, I., Baghy, K. Tumor-specific inhibitory action of decorin on different hepatoma cell lines. Cell Signal. 2019; 62: 109354.
9 Appunni, S.; Anand, V.; Khandelwal, M.; Gupta, N.; Rubens, M., Sharma, A. Small Leucine Rich Proteoglycans (decorin, biglycan and lumican) in cancer. Clin Chim Acta. 2019; 491: 1-7.
Reviewer 2 Report
The study is interesting. I like it overall
Recommendations/comments.
- English writing/editing can be improved. Too many linking words at the start of some sentences, some abbreviations should be defined the first time they appear etc (please see PDF with some comments in case they can help).
- Please tone down your statements as the present findings are interesting but still preliminary in essence.
- Some decimals can be removed from table (see my PDF) too make the Table clearer.
- In the introduction the authors make the case that decorin is a myokine. But actually decorin and other potential exercise-factors were not measured as such in the study (ie, they were apparently measured at baseline*, not after an exercise session). So please rewrite the Ms accordingly where applicable b/c we don’t know where decorin and other potential myokines come from (from which tissue) in the context of the present study. Do the authors have post- acute exercise results on these biomarkers? If they could show that decorin does increase after exercise in the present patients we have a proof of principle to recommend exercise for them. Please see my point #7
- *Please more specification is needed on how/when were serum samples obtained (eg, baseline, early morning after an overnight fast)?
- Sarcopenia (2.7) ‘decreased’ compared to what? Isn’t it ‘lower’? Please be more specific
- An illustrative cartoon with the main study results and with hypotheses of which tissues do release decorin and the rest of molecules to the bloodstream + the potential target tissue of decorin and the other molecules would be of great assistance to the reader. Please be creative and illustrative. As mentioned above, in the context of the present design we don’t know if decorin comes from muscle contractions (ie, is a myokine)
- A flow diagram would be great, to convince the readers that the authors dis not use a convenience sample.
- Finally, in the definition of myokine by the authors, please bear in mind that not all myokines are cytokines. They are small peptides in general, but not necessarily cytokines.
Author Response
To REVIEWER 2,
Thank you very much for your letter regarding our manuscript (jcm-734136). We appreciate your comments, which have helped us to improve our manuscript. In line with your comments, please find below our point-by-point responses.
- English writing/editing can be improved. Too many linking words at the start of some sentences, some abbreviations should be defined the first time they appear etc (please see PDF with some comments in case they can help).
Response: We deeply appreciate for sparing your valuable time to review our manuscript. We have corrected grammatical errors and typos following your suggestions indicated in the attached PDF file.
- Please tone down your statements as the present findings are interesting but still preliminary in essence.
Response: As you suggested, our findings is still preliminary. We have tone down our statement by adding the phrase “Although more research is needed and our data are preliminary in essence” following your suggestion (line 202-203, line 289-291).
- Some decimals can be removed from table (see my PDF) too make the Table clearer.
Response: We deeply appreciate for your careful peer review. We have removed some decimals from table following your suggestion indicated in the attached PDF file.
- In the introduction the authors make the case that decorin is a myokine. But actually decorin and other potential exercise-factors were not measured as such in the study (ie, they were apparently measured at baseline*, not after an exercise session). So please rewrite the Ms accordingly where applicable b/c we don’t know where decorin and other potential myokines come from (from which tissue) in the context of the present study. Do the authors have post- acute exercise results on these biomarkers? If they could show that decorin does increase after exercise in the present patients we have a proof of principle to recommend exercise for them. Please see my point #7
Response: We agree with your comment. As you suggested, decorin is expressed not only in the skeletal muscle, but also in intestinal tissue, cardiac tissue, and adipose tissue [1-4]. In addition, serum decorin level was measured using baseline samples, but not samples after an exercise session. Thus, we added the following sentence: decorin is known to be expressed not only in skeletal muscle [5], but also in various tissues including intestinal tissue, cardiac tissue, and adipose tissue [1-4]. Accordingly, it remains unclear where decorin comes from in the present study (line 256-259). Again, we appreciate your comments, which have helped us to improve our manuscript.
- *Please more specification is needed on how/when were serum samples obtained (eg, baseline, early morning after an overnight fast)?
Response: We apologize for unclear statement. Blood samples were obtained at the baseline in the early morning after an overnight fast. In the revised manuscript, we added the description in the Materials and Methods section (line 116).
- Sarcopenia (2.7) ‘decreased’ compared to what? Isn’t it ‘lower’? Please be more specific
Response: We apologize for unclear statement. In this study, the diagnosis of sarcopenia was based on the Japan Society of Hepatology diagnostic criteria for sarcopenia in patients with liver disease [6]. Patients who showed both a decrease in grip strength (the cut-off value is 26 kg for men and 18 kg for women) and a decrease in skeletal muscle mass (the cut-off value of SMI is 42 cm2/m2 for men and 38 cm2/m2 for women) were diagnosed with sarcopenia. In the revised manuscript, we added the description in the Materials and Methods section (line 111-113).
- An illustrative cartoon with the main study results and with hypotheses of which tissues do release decorin and the rest of molecules to the bloodstream + the potential target tissue of decorin and the other molecules would be of great assistance to the reader. Please be creative and illustrative. As mentioned above, in the context of the present design we don’t know if decorin comes from muscle contractions (ie, is a myokine)
Response: As you suggested, we added an illustrative cartoon for the main study results. We draw the cartoon by paying attention to the context of the present design, that we do not know if decorin comes from muscle contraction and the potential target tissue of decorin (Figure 3, line 266-276).
Figure 3. A scheme for proposed hypothesis of this study. Decorin is expressed in various tissues including skeletal muscle, heart, intestine, and adipocytes. In this study, it remains unclear where decorin come from. Decorin may be associated with cardiopulmonary function, because decorin suppresses the atrophy of cardiomyocytes, myocardial fibrosis, and causes macrophage polarization. In addition, decorin may be associated with prognosis of patients with HCC, because decorin downregulates transforming growth factor-β1, epidermal growth factor receptor, glycogen synthase kinase 3β, and extracellular signal-regulated kinase 1/2, G2/M arrest through phosphorylation of cyclin-dependent kinase 1, Down-regulation of vascular endothelial growth factor A, hypoxia-inducible factor 1-α, and hepatocyte growth factor. Abbreviations: TGF β, transforming growth factor-β1; EGF, epidermal growth factor receptor; GSK3β, glycogen synthase kinase 3β; and ERK, extracellular signal-regulated kinase; VEGF, vascular endothelial growth factor; HIF-1α, hypoxia-inducible factor 1-α; HGF, hepatocyte growth factor.
- A flow diagram would be great, to convince the readers that the authors dis not use a convenience sample.
Response: As you suggested, we have added a flow diagram. and revised Subjects section (2.3) as followings: We registered 339 consecutive patients with HCC between November 2014 and March 2018. Of these patients, 165 patients were excluded because of radiofrequency ablation (n=43), hepatic arterial infusion chemotherapy (n=91), tyrosine-kinase inhibitor (n=23), or radiation (n=8), and the remaining 174 patients HCC underwent TACE were selected. Of the 174 patients HCC underwent TACE, 105 patients were excluded because of hepatic encephalopathy (n=27), HCC rupture (n=17), renal failure (n=7), or lack of data for physical function tests (n=54). Finally, a total of 69 patients with HCC were analyzed in this study (line 84-90, Figure 1).
- Finally, in the definition of myokine by the authors, please bear in mind that not all myokines are cytokines. They are small peptides in general, but not necessarily cytokines.
Response: As you pointed out, not all myokines are cytokines. They are small peptides in general. In the revised manuscript, we have revised the definition of myokine according to your suggestion (line 24).
References
1 Svard, J.; Rost, T.H.; Sommervoll, C.E.N.; Haugen, C.; Gudbrandsen, O.A.; Mellgren, A.E.; Rodahl, E.; Ferno, J.; Dankel, S.N.; Sagen, J.V.; et al. Absence of the proteoglycan decorin reduces glucose tolerance in overfed male mice. Sci Rep. 2019; 9: 4614.
2 Zhao, H.; Xi, H.; Wei, B.; Cai, A.; Wang, T.; Wang, Y.; Zhao, X.; Song, Y., Chen, L. Expression of decorin in intestinal tissues of mice with inflammatory bowel disease and its correlation with autophagy. Exp Ther Med. 2016; 12: 3885-92.
3 Gubbiotti, M.A.; Neill, T.; Frey, H.; Schaefer, L., Iozzo, R.V. Decorin is an autophagy-inducible proteoglycan and is required for proper in vivo autophagy. Matrix Biol. 2015; 48: 14-25.
4 Pohle, T.; Altenburger, M.; Shahin, M.; Konturek, J.W.; Kresse, H., Domschke, W. Expression of decorin and biglycan in rat gastric tissue: effects of ulceration and basic fibroblast growth factor. Scand J Gastroenterol. 2001; 36: 683-9.
5 Kanzleiter, T.; Rath, M.; Gorgens, S.W.; Jensen, J.; Tangen, D.S.; Kolnes, A.J.; Kolnes, K.J.; Lee, S.; Eckel, J.; Schurmann, A.; et al. The myokine decorin is regulated by contraction and involved in muscle hypertrophy. Biochem Biophys Res Commun. 2014; 450: 1089-94.
6 Nishikawa, H.; Shiraki, M.; Hiramatsu, A.; Moriya, K.; Hino, K., Nishiguchi, S. Japan Society of Hepatology guidelines for sarcopenia in liver disease (1st edition): Recommendation from the working group for creation of sarcopenia assessment criteria. Hepatol Res. 2016; 46: 951-63.
Reviewer 3 Report
In the present study Kawaguchi et al performed a retrospective study investigate the impact of serum decorin levels on the physical capacity and prognosis in patients diagnosed with hepatocellular carcinoma. In general this is a straight forward study with an interesting finding.
However, there are some points that need to be addressed. Firstly, while the study of Kanzleiter and colleagues imply that decorin is a "myokine" the data are circumstantial. There is no evidence in this previous paper that the increase in circulating decorin in humans was derived from muscle or indeed the contracting limb. In order to state this with confidence artery-venous balance studies showing net leg release of decorin would have to be performed. This was lacking. Therefore, I would caution the authors about categorically stating that decorin is a myosin without definitive evidence.
Secondly, in this study the decorin levels were associated with prognosis. This is not evidence that elevated decorin would affect prognosis, rather that the two were associated. Therefore, the authors should remove the statement in the conclusion that manipulating decorin is a therapeutic strategy.
Author Response
To REVIEWER 3,
Thank you very much for your letter regarding our manuscript (jcm-734136). We appreciate your comments, which have helped us to improve our manuscript. In line with your comments, please find below our point-by-point responses.
- Firstly, while the study of Kanzleiter and colleagues imply that decorin is a "myokine" the data are circumstantial. There is no evidence in this previous paper that the increase in circulating decorin in humans was derived from muscle or indeed the contracting limb. In order to state this with confidence artery-venous balance studies showing net leg release of decorin would have to be performed. This was lacking. Therefore, I would caution the authors about categorically stating that decorin is a myosin without definitive evidence.
Response: As you pointed out, there is no evidence that the increase in circulating decorin level in humans was derived from muscle or indeed the contracting limb. Accordingly, we have removed the word “myokine” from the title and the abstract (line 2, line 24). In addition, we have revised the Introduction and Discussion sections as following: decorin is known to be expressed not only in skeletal muscle [1], but also in various tissues including intestinal tissue, cardiac tissue, and adipose tissue [2-5]. Accordingly, it remains unclear where decorin comes from in the present study. In addition, we have to be cautious of interpretation of our data. expression of decorin is recently reported to be seen in tumor cell such as glioblastoma and is negatively associated with the overall survival rate of patients with glioblastoma multiforme [6]. Thus, further research is required to investigate the expression of decorin in HCC tissue and a causal relationship between decorin and prognosis of the patients with HCC (line 67-69, line 256-263). Again, we appreciate your comment, which have helped us to improve our manuscript.
- Secondly, in this study the decorin levels were associated with prognosis. This is not evidence that elevated decorin would affect prognosis, rather that the two were associated. Therefore, the authors should remove the statement in the conclusion that manipulating decorin is a therapeutic strategy.
Response: We agree with your comment. Since there is no evidence that elevated decorin would affect prognosis of patients with HCC in this study. Therefore, we removed the statement in the conclusion and revised as following: Although more research is needed and our data are preliminary in essence, these data suggest that decorin may be associated with physical function and prognosis in patients with HCC (line 37-38, line 202-203, line 289-291).
References
1 Kanzleiter, T.; Rath, M.; Gorgens, S.W.; Jensen, J.; Tangen, D.S.; Kolnes, A.J.; Kolnes, K.J.; Lee, S.; Eckel, J.; Schurmann, A.; et al. The myokine decorin is regulated by contraction and involved in muscle hypertrophy. Biochem Biophys Res Commun. 2014; 450: 1089-94.
2 Svard, J.; Rost, T.H.; Sommervoll, C.E.N.; Haugen, C.; Gudbrandsen, O.A.; Mellgren, A.E.; Rodahl, E.; Ferno, J.; Dankel, S.N.; Sagen, J.V.; et al. Absence of the proteoglycan decorin reduces glucose tolerance in overfed male mice. Sci Rep. 2019; 9: 4614.
3 Zhao, H.; Xi, H.; Wei, B.; Cai, A.; Wang, T.; Wang, Y.; Zhao, X.; Song, Y., Chen, L. Expression of decorin in intestinal tissues of mice with inflammatory bowel disease and its correlation with autophagy. Exp Ther Med. 2016; 12: 3885-92.
4 Gubbiotti, M.A.; Neill, T.; Frey, H.; Schaefer, L., Iozzo, R.V. Decorin is an autophagy-inducible proteoglycan and is required for proper in vivo autophagy. Matrix Biol. 2015; 48: 14-25.
5 Pohle, T.; Altenburger, M.; Shahin, M.; Konturek, J.W.; Kresse, H., Domschke, W. Expression of decorin and biglycan in rat gastric tissue: effects of ulceration and basic fibroblast growth factor. Scand J Gastroenterol. 2001; 36: 683-9.
Round 2
Reviewer 1 Report
The manuscript was significantly improved that can be attractive to readers, although the results of study have several inherent limitations.